# Barriers and facilitators to integrating the friendship bench into non-communicable disease care in Malawi: A qualitative study

Steven Mphonda[1], Mwawi Ngoma[1], Katherine Waddell[1]*, Griffin Sansbury[1], Kazione Kulisewa[2]*, Milton Wainberg[3], Mina C. Hosseinipour[1,4], Michael M. Udedi[5], Brian W. Pence[6], Bradley N. Gaynes[7], Melissa A. Stockton[8]

1 University of North Carolina Project-Malawi, Tidziwe Centre, Lilongwe, Malawi, 2 Department of Psychiatry and Mental Health, Kamuzu University of Health Sciences, Blantyre, Malawi, 3 Department of Psychiatry, Columbia University New York, New York, United States of America, 4 Department of Medicine, University of North Carolina at Chapel Hill School of Medicine, Chapel Hill, North Carolina, United States of America, 5 NCDs & Mental Health Unit, Ministry of Health, Lilongwe, Malawi, 6 Epidemiology Department, University of North Carolina at Chapel Hill Gillings School of Global Public Health, Chapel Hill, North Carolina, United States of America, 7 Department of Psychiatry, University of North Carolina at Chapel Hill School of Medicine, Chapel Hill, North Carolina, United States of America, 8 Perelman School of Medicine, University of Pennsylvania Philadelphia, Pennsylvania, United States of America

* kkulisewa@medcol.mw (KW), katywaddell@unc.edu (KW)

## Abstract

Comorbid depression is common among people living with non-communicable diseases (NCDs) in Malawi and is associated with reduced treatment adherence and increased NCD and depression symptom severity. This bidirectional negative relationship between NCDs and depression necessitates an appropriate intervention for addressing depression in those living with NCDs in low resource settings. To assess barriers and facilitators to implementing the Friendship Bench (FB) for depression management as part of NCD care, a convenience sample of 22 participants involved in the development, delivery, or receipt of the FB intervention were recruited for in-depth interviews from two facilities offering the FB. Participants included 12 NCD patients, 8 FB counsellors, and 2 of the original FB developers. Implementation facilitators included an understanding of emotional distress, positive patient-provider relationships, understanding the purpose and benefits of the FB, accessibility of FB, and positive collaborative relationships amongst providers. Major implementation barriers included a lack of mental health literacy among patients, providers, and institutions, a lack of acceptance of the use of mental health interventions, misinterpretations of the scope of practice of the FB counsellor, and limited institutional support for the integration of FB into NCD care. While FB is a promising intervention for addressing depressive symptoms in people with NCDs in Malawi, major implementation barriers may impact its feasibility and acceptability for this patient population. These findings can inform future research on adapting the FB for feasible and acceptable implementation for NCD patients in low resource settings.

**Data availability statement:** Our qualitative data set contains sensitive patient information and is not available for sharing publicly because we do not have ethical approval to share beyond our immediate research team. The Research Ethics Boards who have imposed this restriction are: the University of North Carolina at Chapel Hill (contact: irb_questions@unc.edu); the College of Medicine Research and Ethics Committee (COMREC) (contact: comrec@medcol.mw).

**Funding:** This work was supported by the National Institute of Mental Health of the National Institutes of Health under Award Number (1U19MH113203 to KK, MN, and SM). MAS was additionally supported by (K01MH130226). The content is solely the responsibility of the authors and does not necessarily represent the official views of the National Institutes of Health.The funders had no role in study design, data collection and analysis, decision to publish, or preparation of the manuscript.

**Competing interests:** The authors have declared that no competing interests exist.

## Introduction

Depression is commonly comorbid with chronic ailments, such as non-communicable diseases (NCDs) like diabetes and hypertension, [1–3] as these diseases are associated with greater physical morbidity, disability, increased feelings of hopelessness, and increased levels of stress. [4] The prevalence of depression is 2–4 times higher in those living with NCDs than those living without, presenting a major burden to low- and middle-income countries (LMICs) with limited mental healthcare infrastructure to diagnose and treat depression. [3] Though limited data exists on the prevalence of depression among people living with NCDs in Malawi, a recent study at an NCD clinic in Lilongwe found that 18% of adults with type 2 diabetes screened positive for major depression. [5] The same study found that none of these cases of major depression had been detected by NCD clinicians, [5] posing a public health threat to those living with NCDs. The high prevalence rate of depression among NCD patients [6], combined with the failure of primary care clinicians to diagnose and manage comorbid depression, [2] suggests that there is an urgent need to integrate mental health treatment into NCD care. [7]

Unmanaged depression constitutes a major threat to the health outcomes of those living with NCDs in LMICs. Depression in people living with NCDs is associated with reduced treatment adherence and increased symptom severity, among other NCD-related complications. [1,8] The threat to proper management of NCDs and increased symptom severity is especially problematic for low resource settings like Malawi in which NCDs are a major public health problem that caused 40% of deaths in Malawi in 2019. [9] Without adequate resources and training for medical staff, NCDs will continue to be a serious public health concern in Malawi, [10] especially if comorbid depression is not adequately addressed in this patient population.

Specifically, for those living with hypertension, depression may interfere with blood pressure control and increase the risk of cardiac arrhythmias. [1] For those with diabetes and depression, disease management is complicated by the fact that diabetes and depression share many of the same underlying biological and behavioural mechanisms. [11] The shared behavioural mechanisms are especially problematic as the overlapping behaviours of an inactive lifestyle, inadequate or poor diet, and difficulty sleeping may result in worsening symptoms related to both diabetes and depression. [11] These bidirectional relationships between NCDs and depression necessitate an appropriate intervention for addressing depression in those living with NCDs in Malawi.

Though generic pharmacological aids and clinical resources exist throughout Malawian health systems, healthcare workers have little to no training in psychopharmacology, making psychological interventions advantageous and safer for Malawi. Specifically, studies suggest that psychotherapy interventions utilizing a task-sharing approach in sub-Saharan African settings may be an effective approach for addressing mental health issues in adults. [12] One such intervention, called the Friendship Bench (FB), uses problem-solving therapy to reduce depressive symptoms in adults. [12–14] Originally developed in Zimbabwe, the Friendship Bench intervention consists of six counselling sessions focused on identifying and prioritizing a patient's

problems, developing coping strategies and achievable solutions, and encouraging the patient to take action to address these problems. [13,14] Trained lay healthcare workers provide FB in private settings allowing patients to maintain confidentiality and talk freely with their counsellor, especially in settings like Malawi where mental health is heavily stigmatized. [15,16] This intervention has recently been adapted for use in Malawi to improve depression symptoms in adults living with HIV. [17,18] These studies on the use of the FB in Malawi have suggested its feasibility, acceptability, and efficacy for a variety of patient populations, making it an ideal candidate for addressing the mental healthcare gap for patients living with NCDs and mild depression in Malawi.

This exploratory, qualitative study took place at facilities implementing the FB as part of a larger clinic-randomized control trial – the Sub-Saharan African Regional Partnership (SHARP study). The SHARP study integrated evidence-based depression screening and management, including the FB and measurement-based care for antidepressants, into NCD care. [19,20] Conducted in collaboration with the Malawi Ministry of Health, the trial took place in 10 NCD clinics across the country. [19,20] In this study, we aim to explore the facilitators and barriers to successful implementation of the FB in NCD clinics in Malawi as observed at two SHARP facilities.

## Materials and Methods

### Ethics Statement

We obtained ethical approval from the College of Medicine Research and Ethics Committee (COMREC), approval certificate number P.06/20/3073. All participants who agreed to participate in the study provided written informed consent and received a travel reimbursement equivalent to 5 USD (3,500MK). All participants were given a copy of the consent form or had it read to them (for those unable to read). The information sheets contained information on study procedures, expectations, benefits of participating in the study, possible risks, and risk mitigation.

### Study design

The Friendship Bench (FB) is an evidence-based Problem-Solving Therapy (PST) developed in Zimbabwe, amenable to be delivered by lay persons or non-mental health specialists. [12,14] At our study sites, FB was delivered by trained expert-peer FB counselors to eligible NCD patients with mild depression while eligible patients with moderate to severe depression were treated by trained healthcare workers with antidepressant medication. [19,20] In-depth interviews based on the original CFIR constructs [21] were conducted with patients, FB counselors, and FB developers to explore the barriers and facilitators to successful implementation strategies for the FB, as well as the views on the feasibility, sustainability, and acceptability of the delivery of the FB intervention in 2 of the 10 SHARP sites. The study was conducted approximately 2 years after the sites began offering FB counseling.

### Study setting

The study was conducted at two Malawian NCD clinics at Zomba Central Hospital in Zomba district and Bwaila Hospital in Lilongwe. Zomba Central Hospital is a tertiary level hospital serving Zomba district, with a population of 851,737, as well as the eastern region of Malawi. [22], Bwaila Hospital operates at a secondary care level facility for Lilongwe district covering a population of 2,626,901. [22] The two facilities had been providing the FB intervention delivered by trained peer FB counsellors since 2019.

### Study population and recruitment

Individuals involved in the development, delivery, and receipt of the Friendship Bench, including the two developers of the FB, FB counsellors, and FB patients, were recruited for this study from June 1st to December 31st 2021. FB counsellors were peer counsellors living with an NCD. All participants were aged 18 or older and consented to participate through

written informed consent or had it read to them if participants were unable to read. The SHARP study provided stipends for participant transportation. [19,20]

The interviews were conducted with a convenience sample of patients (n = 12) enrolled in the SHARP study and returning for NCD care. Patients were eligible if they had comorbid depression and had attended at least 1 FB session. Patients at each clinic were approached consecutively until 6 individuals met the inclusion criteria and consented to participate in the study. The number of patients meets the expectations for data saturation while also taking into account resources and time constraints. The study research assistants (RAs) identified eligible participants and invited them to participate.

A convenience sample of FB counselors (n = 8) and developers (n = 2) were also interviewed. Inclusion criteria for FB counselors included a secondary level of education, the ability to read and write, the successful completion of a 10-day training, and the completion of at least 6 FB sessions with depressed patients in NCD clinics. RAs approached the FB providers at both facilities to arrange interviews. Of the 10 FB counsellors involved in the delivery of the intervention, 8 were able to participate in the interview based on availability. The study's principal investigators scheduled interviews with the two individuals involved with the development of the original FB intervention and had experience implementing the FB in primary care settings during their visit to Malawi.

## Data collection process

The researchers developed open-ended interview guides to investigate the barriers and facilitators to successful FB implementation. These guides were tailored to each participant group and contained questions designed to obtain perspectives on the feasibility, acceptability, and sustainability of the FB intervention in NCD care in Malawi. The FB developers were also asked about information on the original FB intervention and their experience in implementation of FB in NCD clinics in Zimbabwe. FB counsellors were asked about the mental health and depression priorities at the facilities and the process of introducing and implementing the FB with probes to investigate barriers and facilitators at the intervention-, individual-, facility-, district- and implementation-process level. FB patients were asked about their understanding of the FB and the barriers and facilitators to their engagement with their counselling sessions.

The interview guides used to collect data from FB developers were in English, while the guides for the FB counsellors and FB patients were developed in English and translated into Chichewa by the researchers using a robust process of forward and backward translation. Following the translation, the guides were pre-tested with 4 FB patients not enrolled in this qualitative study. The team refined the guides using a rapid analysis of these 4 pre-test in-depth-interviews (IDIs). The interview guides were consequently modified through an iterative process before the commencement of data collection until the researchers were confident in the performance.

Data was collected by two research assistants, male and female, trained at the diploma and the graduate level, with over 10 years of experience in qualitative work. The research assistants conducted, and audio recorded the interviews in quiet, private offices within the NCD clinic. All interviews lasted 30–45 minutes.

Though not directly assessed as a part of this study, the feasibility and fidelity of the FB intervention were assessed as part of the larger SHARP study trial. [19,20]

## Data analysis

Guided by the revised CFIR constructs, [21] data was analysed using the content analysis approach. All audiotaped interviews were transcribed verbatim and then translated into English. The researchers reviewed all transcripts against audiotapes to check the accuracy of the transcript. The transcripts were imported into Dedoose version 9.0 [23] for analysis.

A deductive codebook was then developed by the study team to categorize interview responses regarding barriers and facilitators according to the CFIR, organizing the identified multilevel barriers and facilitators within its framework. [21] These codes were primarily designed to assess data within the Outer Setting and Inner Setting domains of the CFIR, such as the needs of the intervention receivers, or the patients (Outer Setting), and the needs of intervention deliverers,

the FB counsellors (Inner Setting). [21,24] Each of the 4 researchers first coded 4 transcripts separately and then collaboratively reviewed the coding through a line-by-line comparison process. This process involved members of the coding team explaining their individual coding application and understanding of each code's definition until they collectively agreed upon a standardized use of the codebook. Consequently, the codebook was updated through an iterative process throughout this double coding process. Three researchers then independently coded the remaining transcripts in Dedoose. [23]

The research team conducted thematic analyses to summarize the barriers and facilitators to the implementation of the FB using memos from the coding process. These memos were developed through the review of the code reports related to the barriers and facilitators of implementation at the patient-, FB counsellor-, and institution and implementation-levels that could be mapped back to the CFIR. These memos were then used to inform the analysis of the qualitative data and to draw conclusions from patients', FB counsellors', and FB developers' feedback.

### Inclusivity in global research

Additional information regarding the ethical, cultural, and scientific considerations specific to inclusivity in global research is included in the Supporting Information (S1 Checklist).

## Results

### Participant characteristics

Participants included FB counsellors (FBC) (n = 8), patients (n = 12), and key developers of FB (n = 2) (Table 1). Of the participant groups asked about their physical comorbidities (providers and patients), hypertension was the most common comorbidity (14/20 participants) and diabetes was the second most common physical comorbidity (6/20 participants) (Table 1). In these participant groups, 2/20 participants had both hypertension and diabetes (Table 1). Patients reported attending an average of 3.5 sessions with the highest number of sessions attended by a patient being 4 sessions (Table 1).

**Table 1. Participant Characteristics.**

| N or Mean range | Providers | Patients | Developers |
|---|---|---|---|
| Total | 8 | 12 | 2 |
| *Gender* | | | |
| Male | 1 | 2 | 0 |
| Female | 7 | 10 | 2 |
| *Average Age* | 55 | 50 | 38 |
| *Employment Status* | | | |
| Employed | 2 | 1 | 2 |
| Unemployed | 6 | 11 | 0 |
| *Marital Status* | | | |
| Married | 6 | 9 | 1 |
| Widow | 2 | 2 | 0 |
| Separated | 0 | 1 | 0 |
| Single | 0 | 0 | 1 |
| *Physical Comorbidity* | | | |
| Hypertension | 6 | 7 | N/a |
| Diabetes | 2 | 3 | N/a |
| Both | 0 | 2 | N/a |
| *Average Sessions Attended* | N/a | 3.5 | N/a |

Identified barriers and facilitators were organized at the 1) patient-level; 2) FB counsellor-level, and 3) institutional- and implementation-level and are described below, including an explanation of how each barrier and facilitator maps to the original CFIR constructs.

## Patient-level barriers

Patient-level barriers to successful implementation of the FB for NCD patients with comorbid depression include a lack of clinical knowledge about mental disorders, specifically depression. This barrier is relevant to the Knowledge and Beliefs about the Intervention construct within the Characteristics of Individuals domain of the CFIR and the Patient Needs and Resources construct within the Outer Setting domain as mental health knowledge, sociocultural values and beliefs may impede patient engagement and participation in psychosocial treatment. [21,24]

FB counsellors reported struggles with engaging patients with NCDs and comorbid depression in the FB because for many patients, the FB intervention was their first introduction to depression as a clinical disorder. Many people think of depression as part of life as "it is meant to be" rather than a mental health disorder. This perception of depression also fuels a lack of acceptance of a depression diagnosis and the need for clinical intervention, further driving patients away from seeking out mental healthcare. This lack of acceptance and diminishment of depression's significance burdened FB counsellors who were often unsure of how to navigate such complex care-seeking situations.

*"It's hard for many people to understand and accept depression, most of them (NCD patients) don't know that depression is an illness own its own, so it was hard to explain to someone that they have depression and the care they needed to be getting." – FB Counsellor*

This unwillingness to engage in care for depression at the patient-level can be explained by the community and society-wide issue of the lack of prioritization of mental health care which also aligns with the Knowledge and Beliefs about the Intervention construct within the Characteristics of Individuals domain of CFIR. [21,24] These sociocultural values and beliefs contribute to stigma surrounding depression which often results in delayed care-seeking and more severe manifestations of depression for patients that are more difficult to address using only the FB.

## Patient-level facilitators

However, some sociocultural values and beliefs also acted as a facilitator of the FB intervention. Many patients reported sharing a common understanding of emotional distress as being harmful to one's ability to live a fulfilling life with an NCD. Additionally, many patients believed that positive therapeutic relationships with FB counsellors may combat this emotional distress inherent to the experience of living with an NCD. This common understanding of emotional distress is a facilitator that maps to the Knowledge and Beliefs about the Intervention construct within the Characteristics of Individuals domain of the CFIR as believing in the benefits of positive therapeutic relationships on emotional distress increases patient engagement and participation in mental healthcare. [21] Despite the lack of acknowledgment of depression as an illness, many patients were able to verbalize that living with an NCD could lead to "retained anger" in one's heart, indicating some recognition of the emotional side effects of NCDs. Additionally, many patients understood that this "retained anger" warranted seeking care as these emotions have the potential to negatively impact one's physical health.

This general understanding of emotional distress associated with NCDs facilitated acceptance of and access to the FB as it increased interest in the counselling program. Understanding the benefits of the FB for mental health, as well as the ability to connect the FB to NCD services, increased participant motivation and session attendance. Satisfaction with the FB increased over time as patients built healthy therapeutic relationships with their FB counsellors while developing positive coping strategies to address depression symptoms. Patients praised the FB counsellors for facilitating welcoming and calming environments in which patients felt "free and comfortable" discussing intimate topics. Facilitating such a safe

environment allowed counsellors to establish rapport with patients and help educate patients on the utility of psychosocial interventions in improving their mental and physical health.

All the facilitators allowed patients to interact positively with their community, enhancing overall understanding and awareness of the FB. In many instances, patients indicated they would recommend the intervention to others in the community despite the negative perceptions of depression. In general, open conversations regarding the FB has the potential to decrease community stigma surrounding mental health treatment and increase mental health awareness and literacy.

## FB counsellor-level barriers

FB counsellor-level barriers to the successful implementation of the FB included inadequate refresher trainings, inconsistent supervision and higher-level support from hospital leadership, difficulty maintaining self-care practices, and competing financial obligations.

The issue of inadequate refresher trainings has implications relevant to the Access to Knowledge and Resources and Available Resources constructs within the Inner Setting domain of the CFIR as these constructs capture both access to knowledge about the intervention and the importance of dedicating training and educational resources for dispersal of knowledge for successful intervention implementation. [21,24] The lack of refresher trainings contributed to a reduced quality of care for patients as counsellors often skipped FB protocol steps and lacked clarity on how to manage sessions. Additionally, NCD providers at Bwaila reportedly lacked adequate training in how to screen patients for depression and therefore were referring many patients without depression to counselling. A developer of the FB speaks on the impact of misinformed FB counsellors and screeners on individuals, communities, and the overall implementation of the intervention:

*"…if we don't have good counsellors to provide the service it can be a barrier [and] it will hinder the progress of the organization…It is not about you in a team, but it is about the client out there trying to help them so people within the team need to know what the mission is." – FB Developer*

The developer acknowledges that on a community and implementation level, unhelpful FB sessions could reflect poorly on the program and unsatisfied patients could spread this information, further discouraging community members from seeking help.

Beyond a lack of necessary refresher trainings to improve FB counsellor confidence and ability to provide the intervention, FB counsellors also sometimes struggled to receive the necessary support from hospital leadership. Though FB counsellors reported satisfaction with collaboration with NCD providers, some counsellors reported feeling discontentment with the level of supervision and support from hospital leadership, as relevant to the Leadership Engagement construct of the Inner Setting domain of CFIR. [21]

*"There is nothing that the leadership of this facility is doing at the Friendship Bench…they just see us and sent us patients"– FB Counsellor*

The reported lack of support from the leadership also relates to the Inner Setting domain's Culture construct. [21] Further, the FB Counsellors reported feeling strongly about the importance of the FB intervention for the patients' health and well-being, a sentiment that FB counsellors felt was not shared by facility leaders, demonstrating incongruence with respect to the Relative Priority construct in the Inner setting. [21,24]

This lack of a mutual understanding among leadership and counsellors regarding the importance of the FB intervention reduced counsellor motivation and took a negative toll on many counsellors' self-care efforts. A developer of the FB emphasized the importance of counsellor self-care saying,

*"An empty oil lamp will not burn so if these delivery agents do not have self-care, they will not be able to deliver the service so it is important that there is continuous self-care and support"* – FB Developer

The analogy comparing the FB counsellor to the oil lamp suggests that these counsellors require consistent self-care and support as fuel to deliver the intervention and provide compassionate services. The shortage of self-care opportunities among FB counsellors affected the counsellors' ability to deliver the FB intervention, with implications for the Other Personal Attributes construct with the Characteristics of Individuals Domain (as well-being is relevant to personal, non-technical attributes of interventionists), the Available Resources construct within the Inner Setting domain, and the Executing construct within the Process domain. [21] Without the adequate resources, including support-related resources and supervision, FB counsellors reported challenges with self-care which hindered their ability to properly execute the implementation of the intervention in some cases.

FB counsellors indirectly accused higher-level staff of negatively impacting the perception of the FB within communities through inadequate facility-based support. Counsellors faced extensive barriers to providing high quality care to their patients which could reduce the credibility of the intervention in the eyes of the public.

*"Like I said earlier if we don't have good counsellors to provide the service it can be a barrier because if I sit on the bench with the client and I offer poor services where is that client going to take that information? To social media? To the research counsel? It will hinder the progress of the organization. So having bad delivery agents can also be a hinderer of progress for the project or intervention."* – FB Developer

Another barrier reported was competing roles for counsellors as many are not officially employed by the NCD facility and require other sources of income. Many FB counsellors reported that their stipend was insufficient and aired grievances regarding the delivery of the stipend. This barrier aligns with the Available Resources and Organizational Incentives and Rewards constructs within the Inner Setting as the organization lacked the necessary financial means to pay and, therefore, counsellors reported insufficient financial incentives to maintain commitment to their roles. [21,24]

The issues outlined by the Available Resources, Relative Priority, Funding, Leadership and Engagement, Culture, and Organizational Incentives and Rewards constructs of the CFIR greatly impacted counsellor motivation and capability to maintain high-level delivery of the FB intervention to their patients. [21]

**FB counsellor-level facilitators**

Key factors in facilitation of the implementation of the FB at the FB counsellor-level include an understanding of the utility of the FB and the impact its perceived usefulness had on increasing counsellor motivation, therapeutic skills, and collaborative working relationships with other healthcare providers. Many of the FB counsellors demonstrated an understanding of the importance of FB in addressing the mental health needs within the community. The counsellors' belief in the utility of the intervention functioned as a key facilitator to implementation and aligns with the Individual Stages of Change and the Knowledge and Beliefs about the Intervention constructs of the Characteristics of Individuals domain, [21,24] as counsellors receptivity of and motivation to deliver the intervention is driven by their personal perceived value of the FB. This understanding of the scope and importance of mental health services among counsellors ultimately facilitated positive attitudes towards and acceptance of the FB, as well as the provision of high-quality care. A developer of the FB also suggested that being knowledgeable of the relationship between NCDs and mental health will allow FB counsellors to better contextualize the FB and tailor services to the unique needs of chronic care patients.

*"So, it's important that those that are going to deliver (the FB) should be able or very equipped on what NCDs are, their effects on the health of the clients also their mental wellbeing of the clients."* – FB Developer

Besides a sufficient knowledge base, developers and FB counsellors also mentioned specific qualities and skills that effective counsellors possessed that help patients overcome psychological challenges. Specifically, the therapeutic skills of active listening, confidentiality, openness, and "sensing [patients'] challenges, thoughts, and feelings as if they were your own" were highlighted. These skills align with the Other Personal Attributes construct within the Characteristics of Individuals domain as they demonstrate that counsellors had the interpersonal competence, knowledge, and skills necessary to adequately fulfil their role. [21,24] These therapeutic skills, along with the initial trainings, provided the counsellors with self-confidence in implementation and confidence in the intervention as a means of helping others, mapping again to the Knowledge and Beliefs about the Intervention construct. [21] FB counsellors reported excitement in their ability to have a tangible and positive influence on the lives of their patients as one counsellor reports that the FB "helps" a patient to "be normal again."

Part of returning to normalcy includes having access to comprehensive care for the NCDs and depression which is facilitated through collaborative working relationships between counsellors and NCD healthcare providers. These collaborative working relationships and communications can be mapped to Networks and Communication construct in the Inner Setting domain of the CFIR, [21] as these relationships facilitated referrals for medications and other interventions that complimented the psychotherapeutic approach to addressing depression in NCD patients, demonstrating a shared culture among healthcare providers and FB counsellors. This shared recipient-centeredness culture focused on the care and support needs of patients, corresponding to the Culture construct of the Inner Setting domain. [21,24]

## Institutional- and implementation-level barriers

Major implementation barriers to the successful delivery of the FB to NCD patients acknowledged by all participants included a low level of mental health literacy within communities and in healthcare institutions, challenges with remote counselling, and a lack of physical space devoted to in-person sessions. Frustrations towards leadership and the lack of resources differentiated by district hospital.

A lack of mental health literacy resulted in limited understanding and education regarding comorbid mental health challenges associated with NCDs, as well as stigma that negatively affected patients, FB counsellors, and NCD providers. One FB Developer commented on the current climate surrounding mental health:

*"Some people do not have knowledge in mental health, they do not really understand and because of the stigma, some people are not willing to listen when we educate them on mental health." – FB Developer*

Because of the combined lack of mental health literacy at the societal and institutional level, NCD providers who were screening patients for depression and making referrals could not adequately fulfil their role of screening patients for depression. This limited understanding of depression, corresponding to the Knowledge and Beliefs about the Intervention construct in the Characteristics of Individuals domain, weakened the capacity of the institution to successfully implement the intervention in some cases, having implications Readiness for Implementation within the Inner Setting. [21] However, participants noted that a lack of training around how to treat patients with mental disorders led some NCD providers to show a lack of "love" and empathy for their patients.

*"But when you go to the clinic, right now when you go outside you will find patients are being shouted at." – Zomba Patient*

Within the CFIR, this apathy from providers can be categorized under the Other Personal Attributes construct in the Characteristics of Individuals domain as the stigma stemming from personal belief systems or values negatively impacted the providers' ability to fulfil their supporting roles in the implementation of the FB intervention. [21]

During COVID-19, FB counsellors began providing phone-based counselling as part of an effort to decongest the healthcare facilities. The switch to remote counselling reduced patient engagement and created significant barriers for implementation. FB counsellors had difficulties reaching patients via phone calls, keeping them engaged for follow-up visits, and developing deep and confidential conversations without having a face-to-face connection. The struggle to maintain engagement during remote sessions due to the switch to remote counselling fits within both the Available Resources construct under the Inner Setting domain and the Adaptability construct in the Intervention Characteristics domain. [21] In some cases, the inability to acquire the necessary resources, such as airtime, greatly limited the benefits of the intervention, suggesting that the intervention struggled to be adapted to the remote setting with the same efficacy as delivering the intervention in the in-person setting.

Limited airtime, or Internet data, posed a challenge for FB counsellors delivering phone-based FB, which created barriers to contacting established patients and connecting with new patients.

*"Toward the end we were told that they would be providing us with airtime to be able to call the participants to conduct the sessions via phone. We have only been given the airtime two times and we have not been able to do follow up visit nor have we been able to talk to the new participants as well. So not having adequate airtime has made us not to be reliable and dedicated to the cause we failed to reach out to them." – Bwaila Peer Counselor*

This issue aligns with the Available Resources construct under the Inner Setting domain as the lack of adequate technological resources required for patient contact affected the ability for the intervention to be delivered according to the structured FB protocol. Ultimately, challenges with airtime caused lapses in communication with patients and reduced utility of the intervention.

*"I will say that in the past we were meeting the needs, that was when we were new here and interact with our clients. But now we are working from home, and we are failing to meet the needs of the facility. When they give the phone numbers of our patients, some can't get through maybe for a week. When you make a call, some will tell you that now I am fine which means you can't continue with your interaction, and this is very difficult to work from home." – Zomba Peer Counselor*

FB counsellors also noted limited space was made available for in-person FB, which they believed reflected poorly on facility leadership. Rooms were often not conducive for sensitive conversations, and sometimes were dually booked, forcing FB counsellors and patients to find new spaces to retain confidentiality. This issue also fits within the Inner Setting construct of Available Resources, as the lack of physical space acted as a significant barrier to fulfil the purpose of the FB counselling sessions. [21,24]

Hospital-specific resource shortages were reported by FB counsellors. Counsellors reported a lack of material resources at Bwaila hospital and delays in receiving financial incentives at Zomba Cenral hospital. Three of the FB counsellors at Zomba Central hospital complained of unpaid stipends and resulting negative attitudes towards the FB and perceptions of the facilities ability to reliably support the counsellors. Both hospital-specific issues reported in the interviews align with the Available Resources and Implementation Climate constructs of the Inner Setting, posing additional challenges to the implementation of the FB. [21]

### Institutional- and implementation-level facilitators

Institutional- and implementation-level facilitators included the availability and financial accessibility of the FB, organizational commitment to the FB, and open communication among leadership and FB counsellors.

Patients expressed satisfaction with the accessibility of the FB as it is a free clinical service, and the FB counsellors were often readily available to assist patients. The accessibility of this free service and the general availability of the

counsellors can be mapped within the Available Resource construct of the Inner Setting domain and the Patient Needs and Resources construct of the Outer Setting domain, [21] as the intervention was delivered in location and manner that met the accessibility and financial needs of the patient population. Patients especially appreciated that FB counsellors were often available, even to call on non-clinic days, which reduced the burden of other clinic barriers for patients such as transport difficulties or long wait times to see the counsellor on the designated clinic day. One FB developer reflected on the accessibility of the intervention:

*"It is an intervention where everyone can access and it is free of charge, which has allowed it to scale up into cities and towns in Zimbabwe. The one important thing is that many lives are being saved." – FB Developer*

Organizational commitment to implementation of the FB was also an important facilitator as it eased the adjustment of introducing additional tasks to existing clinical services and reaffirmed the importance of the intervention and the prioritization of depression care. The prioritization of the implementation of the FB intervention, corresponding to the Relative Priority and Implementation Climate constructs of the Inner Setting domain, [21] was acknowledged by FB counsellors who noted that leadership hired empathetic and empowering FB counsellors and NCD providers with positive attitudes. In most cases, these providers and FB counsellors were able to provide effective depression screening, referrals, and in some cases, psychoeducation.

When it occurred, supervision aided in the delivery of the FB by encouraging NCD providers to continuously screen for depression and providing feedback to FB counsellors on adherence to the FB protocol. In some instances, open communication between FB counsellors and facility leadership who prioritized the FB intervention resulted in better health outcomes for patients and fostered a growth-centered environment, mapping to the Leadership Engagement and Networks and Communication constructs in the Inner Setting domain. [21] However, this open communication was not standard as highlighted in FB counsellor-level barriers to implementation.

## Discussion

Many facilitators and barriers are associated with successful implementation of the FB intervention at the patient-, FB counsellor-, and institutional-levels. The main facilitators associated with successful implementation included a general understanding of emotional distress associated with NCDs (Patient Needs and Resources in the Outer Setting domain), positive therapeutic relationships with FB counsellors and understanding the purpose and benefits of the FB (Patient Needs and Resources in the Outer Setting domain), counsellor motivation (Individual Stages of Change/Knowledge and Beliefs about the Intervention in the Characteristics of Individuals domain), collaborative working relationships among healthcare providers (Networks and Communication in the Inner Setting domain), availability and accessibility of the FB (Available Resources in the Inner Setting domain/Patient Needs and Resources in the Outer Setting domain), and open communication among leadership and FB counsellors (Networks and Communication and Leadership Engagement in the Inner Setting Domain). [21,24] While there is a prevalent lack of mental health literacy at both the community- and provider-levels in Malawi, [25,26] an understanding of the emotional distress related to living with an NCD and the perceived usefulness of the FB helped facilitate positive outcomes from the FB and encouraged NCD and depression treatment adherence in patients. Additionally, patients reported positive relationships with counsellors who had been adequately trained to provide FB counselling services.

Despite these facilitators making the FB accessible to patients, there were many barriers, particularly at the FB counsellor- and institutional-levels to successful implementation. There were some struggles reported in implementation due to a lack of acceptance of mental health interventions, mapped within the External Policies and Incentives construct of the Outer Setting domain of the CFIR. [21,24]

Challenges also arose as higher-level staff failed to see the importance of the FB and did not always prioritize the well-being of FB counsellors (Leadership Engagement and Relative Priority within the Inner Setting domain). [21,24]

These challenges underscored the concern that mistakes made in the implementation of the FB may add to the negative perception of help-seeking behaviours for mental health disorders, a perception that has led to the stigmatization of those in sub-Saharan Africa who attempt to receive mental healthcare. [27] This perception also discourages others in need of mental healthcare from seeking medical help, resulting in a higher prevalence of delayed intervention and more severe manifestations of mental health disorders, [27] both of which have become major public health concerns in Malawi. [2]

These negative mental health outcomes related to a lack of help-seeking behaviours are particularly problematic for those in Malawi living with NCDs. As studies suggest, mental health disorders, particularly depression, are associated with reduced treatment adherence and increased symptom severity in people living with NCDs. [1,8] This reduction in NCD treatment adherence for those with depression is largely due to the behavioural symptoms of depression such as isolation, lack of motivation, increased feelings of hopelessness, and fatigue that decrease healthcare visit attendance. [28] Beyond the problematic behavioural symptoms, those with depression and hypertension also experienced increased complications related to blood pressure and other cardiac complications, increasing the quantity and severity of their NCD-related symptoms. [1] For those with diabetes, depression is a serious threat to proper disease management and is more likely to result in negative physical and mental health outcomes. [11]

Reducing these identified barriers to the implementation of the FB, a depression intervention that has been adapted for use in Malawi, is necessary to improve the overall health and well-being of those living with NCDs in Malawi. To reduce these barriers, education regarding depression and its role in NCD health outcomes is imperative at the community-, patient-, FB counsellor-, and institutional-levels. Improving mental health literacy would encourage patient-level help-seeking behaviours, reduce the burden of education on FB counsellors, and help institutions to prioritize mental health interventions for NCD patients. A greater understanding of the importance of the FB intervention in assisting patients with NCDs may also improve working relationships between FB counsellors and higher-level staff, easing logistical barriers such as booking private rooms or ensuring adequate supervision over counsellors and providers screening patients for depression. For patients with NCDs and comorbid depression, understanding the role of depression in their disease management may help motivate them to seek out mental health interventions and have a better grasp of their own behaviours related to treatment adherence. Having a better understanding of how the behavioural symptoms of depression may be negatively influencing their health behaviours associated with their NCDs may help patients improve their mastery over disease management, reducing the burden on FB counsellors and healthcare providers.

### Limitations

Limitations of this study include sampling and social desirability bias, translation loss, interviewer effects, and site-specific resource disparities between the two clinics. The majority of patients interviewed in this study were female (10/12) which does not accurately represent the predominantly male NCD patient population in Malawi. [2] Additionally, these patients were seeking care in urban healthcare centres which makes the sample less generalizable to the population as a large portion of the population lives and seeks care in rural settings. While efforts such as utilizing private rooms and revising the interview guides based on pre-tests were employed to ensure patients' comfortability and confidentiality, patients may have been subject to social desirability biases. Beyond social desirability bias effects, interviews may have been impacted by interviewer effects such as choices in wording and body language reactions. Prior to the implementation of the intervention, interviewers were trained to use nonjudgmental and standardized language to mitigate interviewer effects. Translation loss, such as a distortion of meaning once content was translated from Chichewa to English for the analysis of the qualitative data, may also have impacted the results of this study despite the use of professional translators and bilingual study team members. Differences in resource availability at the two sites may have also skewed the experience of FB receivers and deliverers.

## Conclusion

This qualitative study aimed to explore the facilitators and barriers to successful implementation of the FB at two NCD clinics in Malawi. In-depth interviews with patients, FB counsellors, and FB developers yielded detailed insights into the patient-, FB counsellor-, institutional- and implementation-level facilitators and barriers of the FB at NCD clinics. Findings from this study suggest that the FB has the potential to be an acceptable depression intervention for patients with NCDs and comorbid depression, noting opportunities to improve cultural appropriateness and the need for further education and training of healthcare facility staff and FB counsellors. The lack of mental health literacy continues to be a major implementation barrier for mental health interventions in Malawi and greatly impacted the successful delivery and receipt of the FB in this study. To improve future mental health interventions, specifically for people living with NCDs in Malawi, public education regarding the impact of depression on NCDs must be prioritized to increase help-seeking behaviours in this population and reduce the predominant barrier of stigma.

## Supporting information

**S1 Checklist.** *Inclusivity in Global Research Questionnaire.* The questionnaire outlines ethical, cultural, and scientific considerations specific to inclusivity in global research.
(DOCX)

## Acknowledgments

We would like to express our gratitude to the participants who took time to participate in the interviews and who generously allowed those interviews to be used in this analysis. We would also like to acknowledge the interviewers, without whom this qualitative study would not have been possible.

## Author contributions

**Conceptualization:** Steven Mphonda, Mwawi Ngoma, Kazione Kulisewa, Milton Wainberg.

**Data curation:** Steven Mphonda, Mwawi Ngoma, Griffin Sansbury, Kazione Kulisewa.

**Project administration:** Milton Wainberg.

**Supervision:** Kazione Kulisewa, Mina C. Hosseinipour, Michael M. Udedi, Brian W. Pence, Bradley N. Gaynes, Melissa A. Stockton.

**Writing – original draft:** Steven Mphonda, Mwawi Ngoma, Katherine Waddell, Griffin Sansbury.

**Writing – review & editing:** Milton Wainberg, Mina C. Hosseinipour, Michael M. Udedi, Bradley N. Gaynes, Melissa A. Stockton.

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
