## [Decision Letter · Decision Letter 0]

19 Jun 2025

PMEN-D-25-00236

Barriers and facilitators to integrating the Friendship Bench into Non-Communicable Disease care in Malawi: a qualitative study

PLOS Mental Health

Dear Dr. Waddell,

Thank you for submitting your manuscript to PLOS Mental Health. After careful consideration, we feel that it has merit but does not fully meet PLOS Mental Health’s publication criteria as it currently stands. Therefore, we invite you to submit a revised version of the manuscript that addresses the points raised during the review process.

We look forward to receiving your revised manuscript.

Kind regards,

Lambert Zixin Li

Academic Editor

PLOS Mental Health

Journal Requirements:

1. Please include a complete copy of PLOS’ questionnaire on inclusivity in global research in your revised manuscript. Our policy for research in this area aims to improve transparency in the reporting of research performed outside of researchers’ own country or community. The policy applies to researchers who have travelled to a different country to conduct research, research with Indigenous populations or their lands, and research on cultural artefacts. The questionnaire can also be requested at the journal’s discretion for any other submissions, even if these conditions are not met.  Please find more information on the policy and a link to download a blank copy of the questionnaire here: https://journals.plos.org/mentalhealth/s/best-practices-in-research-reporting. Please upload a completed version of your questionnaire as Supporting Information when you resubmit your manuscript.

Additional Editor Comments (if provided):

Reviewers' comments:

Reviewer's Responses to Questions

**Comments to the Author**

1. Does this manuscript meet PLOS Mental Health’s publication criteria? Is the manuscript technically sound, and do the data support the conclusions? The manuscript must describe methodologically and ethically rigorous research with conclusions that are appropriately drawn based on the data presented.

Reviewer #1: Yes

Reviewer #2: Yes

2. Has the statistical analysis been performed appropriately and rigorously?

Reviewer #1: No

Reviewer #2: Yes

3. Have the authors made all data underlying the findings in their manuscript fully available (please refer to the Data Availability Statement at the start of the manuscript PDF file)?

Reviewer #1: No

Reviewer #2: Yes

4. Is the manuscript presented in an intelligible fashion and written in standard English?

Reviewer #1: Yes

Reviewer #2: Yes

5. Review Comments to the Author

Reviewer #1: 1. Barriers/facilitators are listed but not mapped rigorously to CFIR domains (e.g., "lack of supervision" should link to Inner Setting to Implementation Climate).

2. Remote counseling challenges are noted (p.16) but not integrated into the analysis. Pandemic disruptions (e.g., staff shortages, resource diversion) likely amplified institutional barriers but are unexplored.

3. Page 8. Insufficient detail on how CFIR constructs were applied. The "iterative codebook refinement" process lacks specifics (e.g., how discrepancies among coders were resolved).

4. Acceptability/feasibility are assessed anecdotally. No structured metrics (e.g., fidelity checks, attendance rates, refusal logs) are reported.

5. Prevalence estimates for depression among NCD patients rely on disparate sources (some not Malawi‐specific) and some are dated (e.g., Udedi 2014). A clearer synthesis of current Malawian data—particularly post‐COVID—would strengthen the rationale.

6. Sampling problem is hard to solve: Convenience sampling of 6 patients per clinic risks selection bias and limits transferability; there is no explanation of how saturation was assessed or reached. And gender imbalance: 10/12 patient participants are female (Results, Table 1), yet the authors note the NCD population is predominantly male. The potential impact of this skew is not discussed.

7. The paper acknowledges sampling and social desirability biases but fails to consider translation loss, interviewer effects, and site‐specific resource disparities (e.g., Bwaila vs. Zomba).

Reviewer #2: GENERAL COMMENT

This is a very interesting paper about an intervention - the Friendship Bench (FB) - that seeks to solve a problem of comorbid depression treatment in primary care. This is a very critical topic since comorbid depression, particularly among patients with NCDs is becoming increasingly common in Sub-Saharan Africa, given the rising burden of NCDs in the region. The authors give a good background and describe a detailed methodology of their research, followed by valid results, a discussion, and a corresponding conclusion. However, the following should be considered to improve the paper:

MAJOR ISSUES

1. I appreciate the effort to describe the methodology in detail, including various actors involved in the intervention and how they were recruited. However, the intervention itself is still not very clear. Even if (and it is not clear yet) the intervention is described elsewhere in another paper, I recommend a section in the introduction of the paper, even if it is just a paragraph or two, that describes or summarizes the FB intervention to aid the reader.

2. The results section of the paper is comprehensive, but it could do with restructuring to aid the reader's understanding. At the moment, the results are structured into patient-level factors, FB counsellor-level factors, etc., but the specific factors under each, whether barriers or facilitators, can be organized further as subsections. If possible, kindly situate the results within a conceptual or theoretical framework.

MINOR ISSUES

3. Beside using the inclusion criteria, could you kindly explain what informed the number of FB users, counsellors, and developers that were interviewed? Was it data saturation or something else?

6. PLOS authors have the option to publish the peer review history of their article (what does this mean?). If published, this will include your full peer review and any attached files.

**Do you want your identity to be public for this peer review?** For information about this choice, including consent withdrawal, please see our Privacy Policy.

Reviewer #1: **Yes: **Peilin Yang

Reviewer #2: No

---

## [Editor Report · Decision Letter 1]

20 Aug 2025

Barriers and facilitators to integrating the Friendship Bench into Non-Communicable Disease care in Malawi: a qualitative study

PMEN-D-25-00236R1

Dear Ms. Waddell,

We are pleased to inform you that your manuscript 'Barriers and facilitators to integrating the Friendship Bench into Non-Communicable Disease care in Malawi: a qualitative study' has been provisionally accepted for publication in PLOS Mental Health.

Best regards,

Lambert Zixin Li, Ph.D.

Academic Editor

PLOS Mental Health

Thank you for thoroughly addressing all reviewer comments. I have carefully reviewed the revised manuscript, your response letter, and the highlighted changes. I find that the paper is of publishable quality.